# Design, Fabrication and Analysis for Al$_2$O$_3$/Ti/Al$_2$O$_3$ Colored Solar Selective Absorbers for Building Applications

**Fu-Der Lai *** and **Wen-Yang Li**

Institute of Photonics Engineering, National Kaohsiung University of Science and Technology, Kaohsiung 82445, Taiwan; winyang0149@gmail.com
* Correspondence: fdlai@nkust.edu.tw

**Abstract:** A good solar selective absorber (SSA) used in building applications must have good color brightness rendering, an excellent photo–thermal conversion efficiency (PTCE) and a long life. The optical thin film design plays an important role before the coating of the optical thin films. In this study, for fabricating a good colored SSA (CSSA), the optical properties and color distribution of Al$_2$O$_3$/Ti/Al$_2$O$_3$ SSA were calculated to obtain the best design. It was found that the color distribution of Al$_2$O$_3$/Ti/Al$_2$O$_3$ SSA with a PTCE in excess of 90% was wide and included all colors, such as red, orange, yellow, green, blue, purple and white. Five colored Al$_2$O$_3$/Ti/Al$_2$O$_3$ solar selective absorbers were designed and fabricated and their performances were analyzed. The best PTCE of a fabricated sample and its thermal emittance efficiency were 93.2% and 7.1%, respectively. The peak values of the measured optical reflectance in the visible region for the five fabricated CSSAs were all greater than 10%. Their lifetime could be very long because the temperature difference between thermal stability of more than 450 °C and the working temperature of less than 150 °C was up to 300 °C. After annealing at 550 °C, the Al$_2$O$_3$–Ti system formed a compound structure of AlTiO$_5$. The Al$_2$O$_3$/Ti/Al$_2$O$_3$ film performances, including the optical properties in the wavelength range of 0.25 to 25 μm, structure, morphology, adhesion, cross-sectional images, AFM image, PTCE, thermal emittance efficiency and thermal stability, were measured and analyzed in detail, and met the characteristic requirements of colored solar absorber films.

**Keywords:** color brightness rendering; chromaticity; colored solar selective absorber; optical thin film design

## 1. Introduction

The optical thin film design plays an important role in coating technologies for the optical thin films. The results of the optical thin film design for the optical properties of the optical thin films can be very close to the results of the coating [1]. Therefore, especially in multi-layer films, the results of optical thin film design can easily and immediately let the designers know whether it meets the designer's optical characteristic requirements. Therefore, optical thin film design has been widely used before coating.

The incorporation of the concept of solar power into housing design includes solar ventilation for buildings. Solar selective absorbers (SSA) are used in the structure of the external walls or roof of a building or dome of a building stairwell to convert sunlight into heat energy to force convection, drawing cooler air into the building, which saves on electricity for air-conditioning [2]. In order to maximize the absorption of solar energy, the surface of the collector absorber is currently black or blue [3–16]. However, the necessity of using black or blue SSAs on building roofs and facades limits their aesthetic appeal and makes architectural integration into the buildings more difficult. Survey results indicate that 85% of architects would prefer colored SSAs (CSSAs) rather than black or blue ones. The development of a CSSA layer material would allow for more diverse designs for easier integration into building plans [17]. Therefore, CSSAs used in building applications

must have good visible brightness i.e., good color brightness rendering. When the peak value of the optical reflectance in the visible region must be greater than 10%, CSSAs will have good color brightness rendering. The better the photo–thermal conversion efficiency (PTCE) of the solar selective absorber layer, the better the performance of the solar heat energy collector. Multilayer films [18–27] for SSA applications have been studied, but very few multilayer films for CSSA [21–27] have been developed. Multilayer coatings of $Cu/Ti/SiO_2/Ti/TiO_2/SiO_2$, colored absorbers with a $TiN_xO_y$ absorbing layer and a $TiO_2/Si_3N_4/SiO_2$ dielectric stack, a double cermet layer Al–AlN solar absorber coating and a five-layered quarterwave stack (optical model: air//HLHLH//glass//HLHLH//air) were elaborately designed for the colored solar selective absorbers. A multilayer film could be the easiest and best structure for a good CSSA. The relationship between the reflection angle and optical reflection spectrum was discussed by A. Schuler et al. To test for thermal stability, some coatings were thermally stable up to 300 °C in air. The absorber lifetime is also important [28]. The better the temperature difference between the thermal stability and working temperature is, the longer the lifetime of the CSSA. In Europe, for example, the lifetime of new buildings must be more than 200 years, requiring the fabrication of colored solar selective absorbers with good color brightness rendering, high PTCE as well as a long lifetime.

In this study, the optical properties and the relationship between the chromaticity distribution and PTCE as well as the layer thickness in $Al_2O_3/Ti/Al_2O_3$ films will be calculated and analyzed in detail by using the CUDA C rapid parallel computation technology. Five CSSAs meeting the requirements to produce a wide color gamut were designed and fabricated. Their visible optical reflectances were all greater than 10%. The performances (including optical properties in the wavelength range of 0.25 to 25 μm, structure, morphology, adhesion, cross-sectional images, AFM image, PTCE, thermal emittance efficiency and thermal stability) of the fabricated films were measured and met the characteristic requirements of a CSSA for building applications. The service lifetime of the fabricated $Al_2O_3/Ti/Al_2O_3$ CSSA was analyzed. The changes in the performances of the fabricated CSSA before and after high-temperature heat treatments were also analyzed.

## 2. Experimental Details

### 2.1. Preparation and Characterization

#### 2.1.1. Preparation of $Al_2O_3/Ti/Al_2O_3$ Solar Selective Absorbers

The $Al_2O_3/Ti/Al_2O_3$ SSAs were deposited on the mirror-like surface of stainless steel SS304 substrates using an unbalanced reactive magnetron sputtering system with a cryopump. The temperature of a sample in the coating chamber was controlled by an infrared heater at 350 °C. The substrates were cleaned in an ultrasonic bath by a series of processes [1,6]. The target materials for the $Al_2O_3$ and Ti were aluminum (99.995 %) and titanium (99.995 %.), respectively. Prior to deposition, the targets were pre-sputtered for 20 min in Ar at a pressure of 10 mTorr to remove any $Al_xO_{1-x}$ and Ti contaminants, and then for 20 min under the deposition parameters to obtain pure $Al_2O_3$ and pure Ti films. The $Al_2O_3$ films were deposited under the following parameters: 3 mTorr pressure, 30 sccm Ar flow rate and 12 sccm $O_2$ flow rate, and 300 W RF sputtering power from an Al target. The Ti film was deposited under the following parameters: 3 mTorr pressure, 30 sccm Ar flow rate and 150W DC sputtering power from a Ti target.

#### 2.1.2. Characterization of Fabricated Thin Films

All color photos were taken by a camera under the sun's rays from 3:00 pm to 4:00 pm in September. The top-view image and cross-sectional image of the films were measured by field emission scanning electron microscopy (FESEM). The top-view image was also measured by an Atomic Force Microscope (AFM). The reflectance spectra in the wavelength ranges of 250–2000 nm and 400– 4000 cm$^{-1}$ (i.e., 2.5– 25 μm) were measured by an optical spectrometer (V-670, ISN-723) and FTIR (Thermo Scientific, Nicolet iS 5), respectively. The refractive index $n$ and extinction coefficient $k$ of the $Al_2O_3$ and Ti films were measured

by a spectroscopic ellipsometer (J. A. Woollam, M-2000 DI). The adhesion between the multilayer film and the substrate was analyzed using the American Society of Mechanical Engineers' crosshatch tape testing method. X-ray diffraction (XRD) was used to examine the structure of the films.

*2.2. Optical Thin Film Design for Solar Selective Absorber Films using the CUDA C Parallel Computation Technology*

To design a good CSSA for building applications, a multilayer film could be the easiest and best structure. Between the upper and lower boundaries in the multilayer films, solar light will be multiply reflected. This reflected light causes the film interference phenomenon (i.e., the reflected light is enhanced or reduced in any one wavelength.) Therefore, it is necessary to process a very large amount of data to calculate the optical reflectance and analyze the chromaticity distribution and PTCE for a good CSSA.

To speed up the computation time for processing large data sets, a data-parallel programming model is used. In this study, the CUDA C rapid parallel computation technology was used in the optical thin film design software with the color distribution analysis program.

2.2.1. Optical Property Simulation of $Al_2O_3/Ti/Al_2O_3$ Multilayer Optical Films

A Ti film with a thickness of 132 nm was used as the adhesion layer and the substrate.

For a multilayer film with m layers, at normal incidence, the characteristic matrix is of the form [6,29]

$$\begin{bmatrix} B \\ C \end{bmatrix} = \prod_{r=1}^{m} \begin{bmatrix} cos_{\alpha r} & \frac{isin_{\alpha r}}{N_r} \\ N_r sin_{\alpha r} & cos_{\alpha r} \end{bmatrix} \begin{bmatrix} 1 \\ N_s \end{bmatrix},$$ 
(1)

$$N_r = n_r - ik_r,$$
(2)

$$\alpha_r = \frac{2\pi}{\lambda} N_r d_r$$
(3)

where $N_r$ represents the optical constants of layer r, $n_r$ is the refractive index of layer r, $k_r$ is the extinction coefficient of layer r, $d_r$ is the thickness of layer r and $N_s$ is the optical constants of the Ti substrate.

The reflectance R is calculated as follows [6]:

$$R = \left( \frac{\eta_0 B - C}{\eta_0 B + C} \right) \left( \frac{\eta_0 B - C}{\eta_0 B + C} \right)^*$$
(4)

where the $\eta_0$ in air is set to be 1.

Using Equations (1)–(4), the optical reflectance of the $Al_2O_3/Ti/Al_2O_3$ multilayer film on the Ti film/substrate is simulated.

2.2.2. Calculation of PTCEs and Thermal Emittance Efficiency for Fabricated $Al_2O_3/Ti/Al_2O_3$ Multilayer Optical Films

Calculation of PTCE:

By fitting the reflectance data to the spectral power density in the standard AM 1.5 spectrum [5], the PTCE was calculated:

$$PTCE = 1 - \frac{\int_{0.25}^{2.0} R(\lambda) \cdot SD(\lambda) d\lambda}{\int_{0.25}^{2.0} SD(\lambda) d\lambda}$$
(5)

where $R(\lambda)$ and $SD(\lambda)$ are, respectively, the measured optical reflectance and the spectral power density of AM1.5 at the wavelength of $\lambda$.

Calculation of thermal emittance efficiency ($\varepsilon$):

Assuming that the surface temperature of a CSSA is 150°C at working, its $\varepsilon$ is calculated:

$$\varepsilon \cdot = \frac{\int\limits_{2.5}^{25} a(\lambda) \cdot TE(\lambda) d\lambda}{\int\limits_{2.5}^{25} TE(\lambda) d\lambda} \tag{6}$$

$$a(\lambda) = 1 - R(\lambda) \tag{7}$$

where $a(\lambda)$, $R(\lambda)$ and $TE(\lambda)$ are, respectively, the absorption, the measured IR reflectance and the spectral power density of the ideal thermal emittance at 150 °C at the wavelength of $\lambda$.

### 2.2.3. Analysis of Chromaticity for Fabricated $Al_2O_3/Ti/Al_2O_3$ Multilayer Films

The tristimulus values for a color with a spectral power distribution $I(\lambda)$ [27] are given in terms of the standard observer by

$$X = \int_{380}^{780} I(\lambda)\overline{x}(\lambda)d\lambda, \tag{8}$$

$$Y = \int_{380}^{780} I(\lambda)\overline{y}(\lambda)d\lambda \tag{9}$$

$$Z = \int_{380}^{780} I(\lambda)\overline{z}(\lambda)d\lambda \tag{10}$$

where $\overline{x}(\lambda), \overline{y}(\lambda)$ and $\overline{z}(\lambda)$ are the CIE 1931 color-matching functions, and $I(\lambda) = D65(\lambda)R(\lambda)$. D65 is the CIE Standard Illuminant D65.

The chromaticity of a color can be specified by the two derived parameters $x$ and $y$, as follows:

$$x = \frac{Y}{X + Y + Z'} \tag{11}$$

$$y = \frac{Y}{X + Y + Z'} \tag{12}$$

where $x$ and $y$ are the values obtained by translating the measured optical reflectance spectra to the CIE1931 chromaticity coordinates.

## 3. Results and Discussion

### 3.1. Calculation for Chromaticity of the $Al_2O_3/Ti/Al_2O_3$ Solar Selective Absorbers

The parameters of the layer thicknesses for the $Al_2O_3/Ti/Al_2O_3$ solar selective absorbers were: thickness of the first layer (top $Al_2O_3$ layer) from 10 to 200 nm, thickness of the second layer (Ti layer) from 1 to 30 nm and thickness of the third layer (bottom $Al_2O_3$ layer) from 10 to 200 nm. The substrate material was Ti.

The n and k for fabricated Ti and $Al_2O_3$ films were measured by an optical spectrometer and are shown in Figure 1a,b, respectively. The simple experimental schematic for the absorber is shown in Figure 1c. The $Al_2O_3$ layer is a transparent layer without any absorption function. The middle Ti layer is an absorption layer. The Ti substrate has two functions: reflection and absorption. At the interfaces of 1 to 4 in Figure 1c, incident light will be partially refracted and partially reflected. Therefore, the absorber is the $Al_2O_3/Ti/Al_2O_3/Ti$ film.

Using Equations (1)–(3) and 8 to 12, the chromaticity of the $Al_2O_3/Ti/Al_2O_3$ films can be found. Figure 1d–e shows the changes in the chromaticity distribution of the $Al_2O_3/Ti/Al_2O_3$ films in the layer thicknesses of the top $Al_2O_3$ layer/Ti layer/bottom $Al_2O_3$ layer: Figure 1d 55/20/X and Figure 1e 70/14/X, respectively. X (the thickness of a bottom $Al_2O_3$ layer) ranged from 40 to 95 nm. The unit of the layer thickness was nm. From Figure 1d, it can be found that the changes in the chromaticity in the CIE 1931 chromaticity

coordinates were linear when the layer thicknesses of the top $Al_2O_3$ layer/Ti layer/bottom $Al_2O_3$ layer were 55/20/X, respectively, and the color changed from light pink to dark orange. As shown in Figure 1e, it was found that the color distribution of $Al_2O_3$/Ti/$Al_2O_3$ multilayer films was wide and close to all light colors, such as light red, light orange, light yellow, light green, light blue, light purple and white. From Figure 1d–e, it is known that any color can easily be designed, but these multilayer films may not have the best PTCEs and colors.

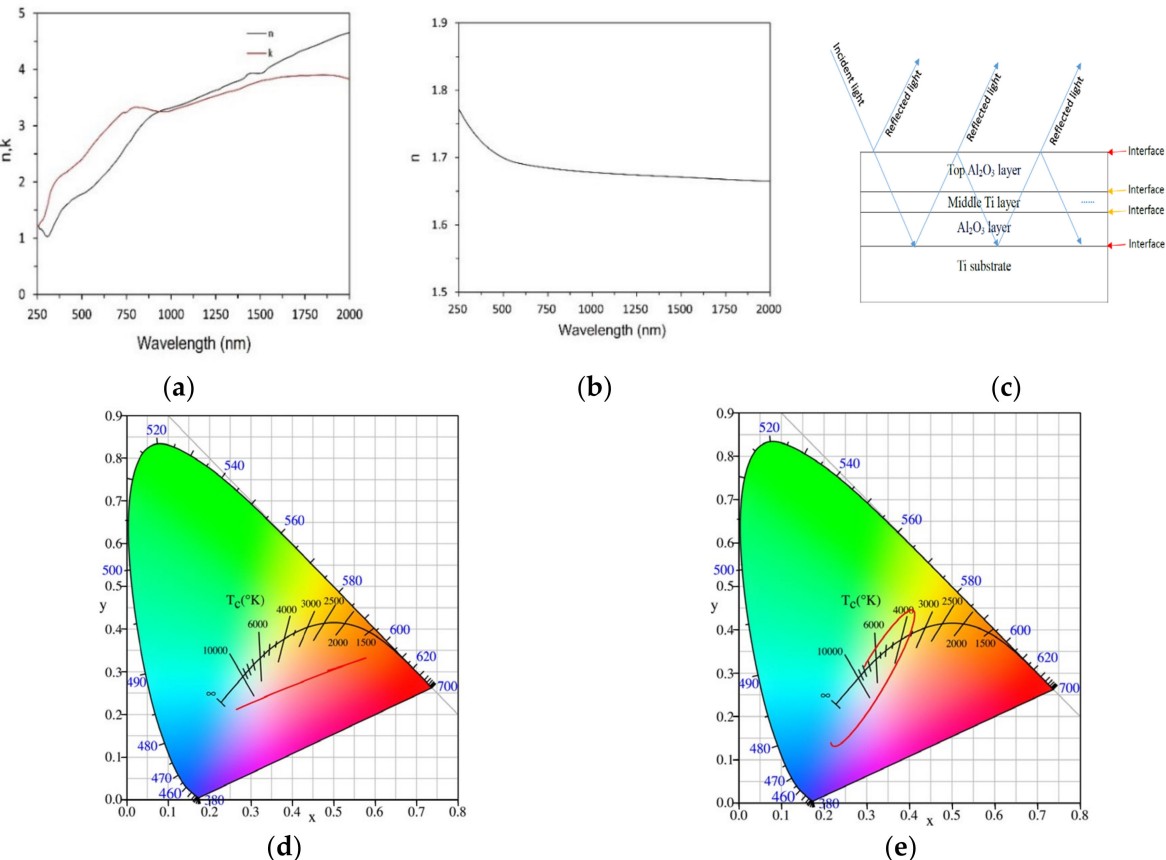

**Figure 1.** (**a**) n and k for Ti and (**b**) n for $Al_2O_3$ films with k = 0. (**c**) Simple experimental schematic for the absorber. Color distribution of $Al_2O_3$/Ti/$Al_2O_3$ multilayer stacks with layer thicknesses of (**d**) 55/20/X and (**e**) 70/14/X. X is the thickness of the third layer ($Al_2O_3$). The layer thickness unit is nm.

### 3.1.1. Calculation of the Chromaticity Distribution for a PTCE of More Than 90%

Using Equations (1)–(7), the PTCE of the $Al_2O_3$/Ti/$Al_2O_3$ films can be found. Figure 2a shows the chromaticity (x,y) of the $Al_2O_3$/Ti/$Al_2O_3$ solar selective absorbers with a PTCE of more than 90% and chromaticity coordinates. The color distribution of these CSSAs is wide and includes all colors, such as red, orange, yellow, green, blue, purple and white.

### 3.1.2. Design of Five Colored Solar Selective Absorbers to Meet the Wide Color Gamut

The design developed to find a wider color distribution shows that the chromaticity (x,y) distribution was wider when the thickness of the second layer was 15 nm, as shown in Figure 2b, than that for other layer thicknesses, such as 18 and 19 nm, as shown in Figure 2c. Thus, there were five designed points of the wider color gamut, respectively, marked as a, b, c, d and e in the chromaticity coordinate diagram of Figure 2b. To have good color brightness rendering, these five designed CSSAs must have peak values of visible light reflectance greater than 10%.

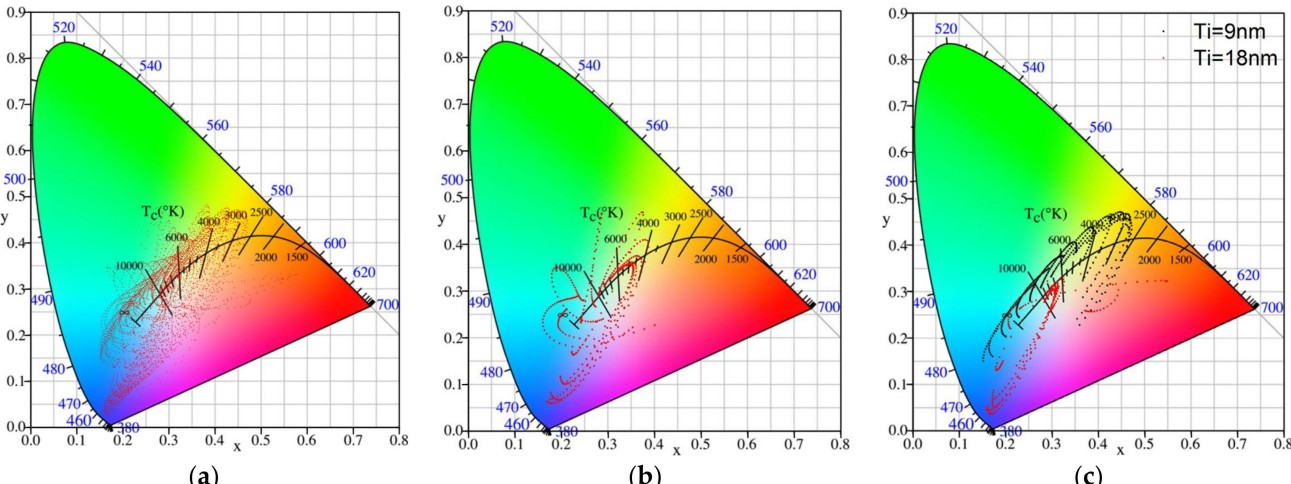

**Figure 2.** (**a**) Chromaticity (x,y) of the Al2O3/Ti/Al2O3/Ti multilayer stacks with a PTCE of more than 90% showing the chromaticity coordinates and the chromaticity ($x, y$) distribution when the second layer has thicknesses of (**b**) 15, (**c**) 9 and 18 nm. Five designed points for a wider color gamut are also shown in the chromaticity coordinate diagram in (**b**).

### 3.1.3. Design of Five Colored Solar Selective Absorbers to Meet the Wide Color Gamut

The design developed to find a wider color distribution shows that the chromaticity (x,y) distribution was wider when the thickness of the second layer was 15 nm, as shown in Figure 2b, than that for other layer thicknesses, such as 18 and 19 nm, as shown in Figure 2c.

CSSAs used in building applications must have good color brightness rendering, an excellent PTCE and a long life. However, good color brightness rendering and excellent PTCE seem to be in conflict, so it is necessary to achieve these two advantages through good design. Thus, there were designed five points for a wider color gamut marked to be a, b, c, d and e, as shown in the chromaticity coordinate diagram of Figure 2b.

### 3.2. Fabrications and Measurements of $Al_2O_3/Ti/Al_2O_3$ Films to Meet the Characteristic Requirements of the Solar Selective Absorber

### 3.2.1. Fabrication and Measurement of Five Colored Solar Selective Absorbers to Meet the Wide Color Gamut

The optical reflectance spectra and color photos of the five fabricated $Al_2O_3/Ti/Al_2O_3$ SSAs corresponding to the five points in Figure 2b are shown in Figure 3a–e. The layer thicknesses of $Al_2O_3/Ti/Al_2O_3$ for samples a to e were, respectively, 45/15/50, 65/15/70, 60/15/69, 55/15/69 and 50/19/58 nm. Figure 3e also shows the simulated optical reflectance. It can be found that the results of the optical properties for the designed film were very close to the measured results for the coated films. Therefore, the structures of the multilayer for the coated films were close to those for the designed films. Sample e with blue color had a better measured PTCE of 93.2%. The peak values of the measured optical reflectance in the visible region for the five fabricated CSSA were all greater than 10%; therefore, the appearance of these colors in the sunlight was very clear and bright (i.e., these five fabricated CSSA all had good color brightness rendering.) Additionally, their measured PTCEs were all greater than 90.5%. Figure 3f shows the IR reflectance spectra of fabricated sample e. Assuming that the surface temperature of the CSSA is 150 °C at working, its thermal emittance efficiency ($\varepsilon$) is 7.1%. Table 1 illustrates the layer thicknesses of $Al_2O_3/Ti/Al_2O_3$, chromaticity coordinates, PTCEs and the $\varepsilon$s of the fabricated $Al_2O_3/Ti/Al_2O_3$ solar selective absorbers in Figure 3a–e.

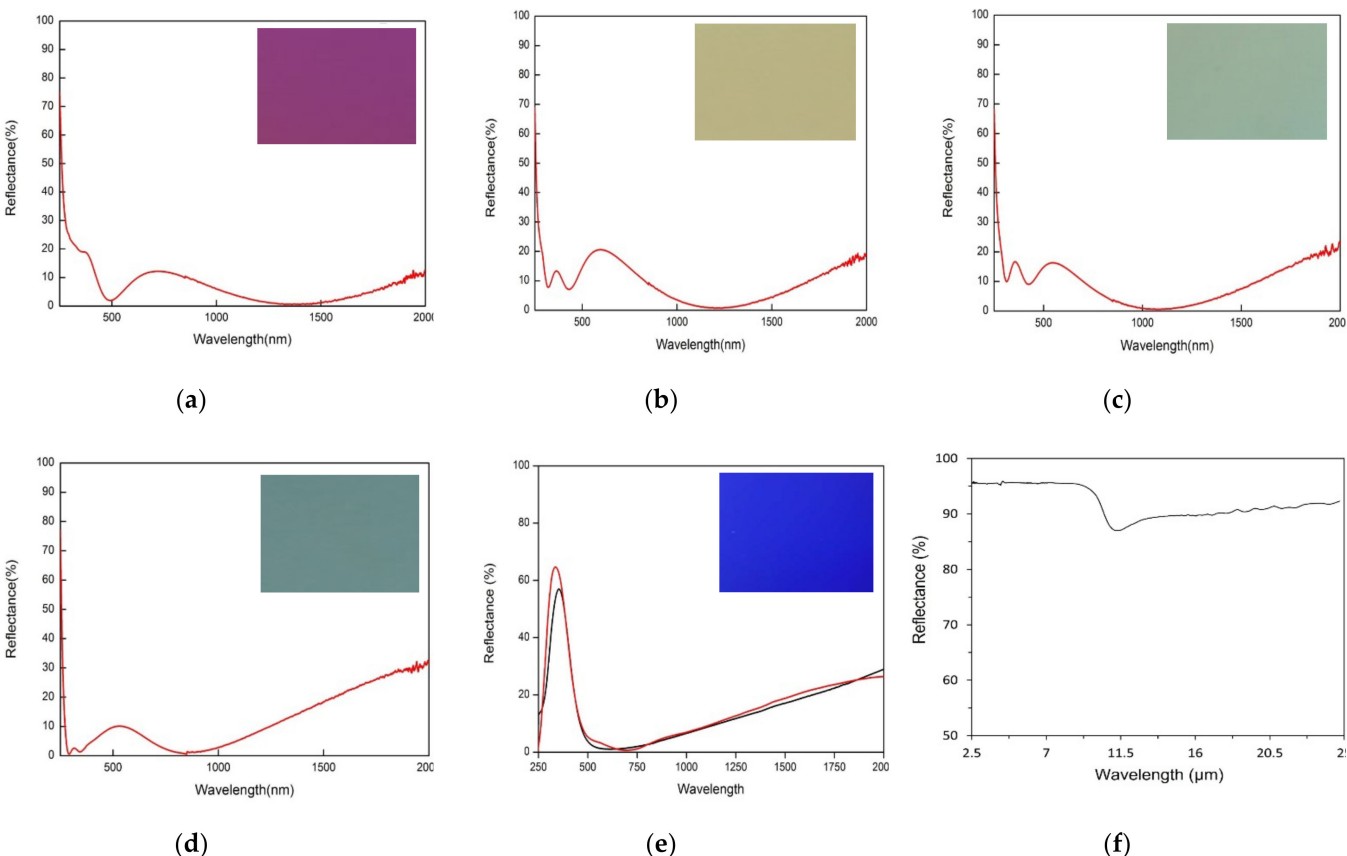

**Figure 3.** Optical reflectance spectra and color photos of the five fabricated samples (**a**)–(**e**) corresponding to the five points in Figure 2b. (**f**) Measured reflectance spectra for fabricated sample (**b**).

**Table 1.** Layer thicknesses of $Al_2O_3/Ti/Al_2O_3$, chromaticity coordinates, PTCEs and $\varepsilon$s of fabricated samples (a)–(e) in Figure 3.

| Sample | Layer Thickness of $Al_2O_3/Ti/Al_2O_3$ (nm) | Chromaticity Coordinate | PTCE (%) | $\varepsilon$ (%) |
|---|---|---|---|---|
| a | 45/15/50 | (0.410, 0.227) | 92.3 | 7.6 |
| b | 65/15/70 | (0.375, 0.467) | 90.5 | 9.1 |
| c | 60/15/69 | (0.286, 0.469) | 91.1 | 8.5 |
| d | 55/15/69 | (0.184, 0.348) | 93.0 | 7.3 |
| e | 50/19/58 | (0.168, 0.064) | 93.2 | 7.1 |

3.2.2. Performance Measurements and Analysis of Fabricated $Al_2O_3/Ti/Al_2O_3$ Solar Selective Absorbers

For real-world applications, good adhesion between the $Al_2O_3/Ti/Al_2O_3$ solar selective absorbers and the substrate is required. In this investigation, the adhesive tape ASTM Crosshatch tape testing method was used. The $Al_2O_3/Ti/Al_2O_3$ CSSAs all passed the adhesion tests. Figure 4a,c, respectively, show a top view and a cross-sectional image of the $Al_2O_3/Ti/Al_2O_3$ solar selective absorber on the substrate measured by FESEM, and Figure 4b shows an AFM image. The surface of the film was slightly rough and the crystal grains in the multilayer stack could be clearly observed. Therefore, due to this slightly rough surface, the measured PTCE of 93.2% of sample e with a blue color was more than the simulated PTCE of 93.0% (its surface is assumed to be very flat.). After annealing three times for 16 h, thermal stability tests were performed on the $Al_2O_3/Ti/Al_2O_3$ CSSA at a temperature range from 150 to 550 °C. There were no changes in their performances, as measured by the FESEM, X-ray and optical reflectance tests, at temperatures of no more than 450 °C. Therefore, in this study, the defined temperature of thermal stability of the

solar selective absorber was 450 °C. Assuming that the temperature of the $Al_2O_3/Ti/Al_2O_3$ CSSA will be less than 150 °C at working, their lifetime could be very long, because the temperature difference between the thermal stability of more than 450 °C and at working was up to 300 °C. Figure 4d shows a top-view image of the $Al_2O_3/Ti/Al_2O_3$ CSSA after annealing three times for 16 h at 550 °C measured by FESEM. It can be found that there were nano-crystal particles of less than 100 nm on the surface. Figure 4e shows the optical properties before and after annealing one and three times for 16 h at 550 °C. It can be found that the optical properties and the color all exhibited small changes after annealing one and three times for 16 h at 550 °C. Therefore, the films nearly changed to be stable after annealing at 550 °C. Figure 4f shows the X-ray diffraction patterns of the solar selective absorbers before and after annealing three times for 16 h at 550 °C. Before annealing, there was only one preferred orientation of (210) in the $Al_2O_3$ layer, but the Ti layer was just crystallized or amorphous. This could be due to the high-density plasma and the sample temperature of 350 °C that caused some crystallinity in the $Al_2O_3$ layer. After annealing, the (210) orientation was transferred to (224) and (512) in the $Al_2O_3$ layer and the Ti layer was well crystallized with (201). It is believed that, after annealing at 550 °C, the $Al_2O_3/Ti/Al_2O_3$ films will have a compound $AlTiO_5$, Ti and $Al_2O_3$, rather than a $TiO_2$. $TiO_2$ did not exist (meaning that the Ti layer had not been oxidized). Therefore, it could be seen that the fabricated $Al_2O_3$ layer was very dense. The $Al_2O_3$–Ti system formed a compound of $AlTiO_5$ (meaning that Ti diffused into $Al_2O_3$ layer to form a compound structure of $AlTiO_5$ and the resulting $AlTiO_5$ layer was highly dense with no pinholes). This is why their optical properties, colors, and PTCEs all only slightly changed after one and three times for 16 h at 550 °C. Therefore, these $Al_2O_3/Ti/Al_2O_3$ CSSAs had good thermal stability. Figure 4e also shows a simulated spectrum of the annealed specimen. The multilayer structure of the specimen changed from $Al_2O_3/Ti/Al_2O_3$ to $Al_2O_3/AlTiO_5/Ti/Al_2O_3$. The multilayer thickness of the specimen changed from 70/15/90 to 65/14/10.5/90 nm. Therefore, the Ti thickness changed from 16 to 10.5 nm and the $AlTiO_5$ layer grew to have a thickness of 14 nm. Additionally, the PTCE changed from 91.9% to 91.3%. The adhesion between the $Al_2O_3/AlTiO_5/Ti/Al_2O_3$ CSSA and the substrate was also very good after annealing.

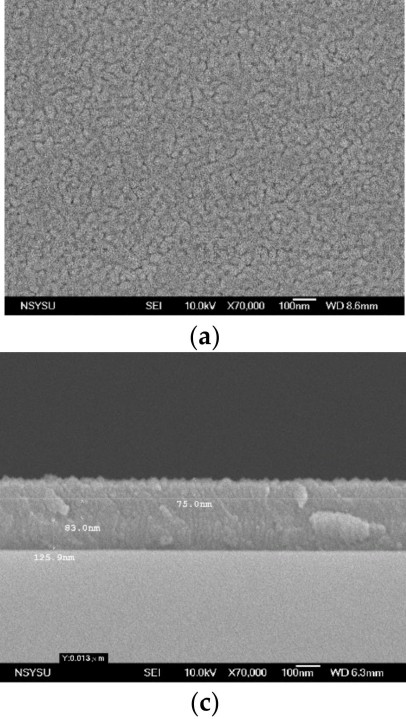

(a)

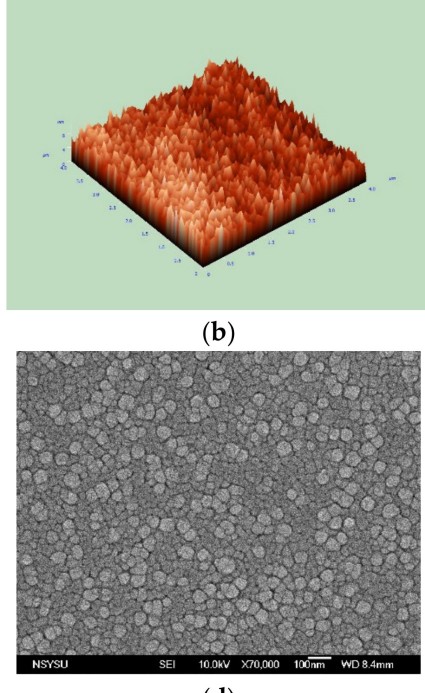

(b)

(c)

(d)

**Figure 4.** *Cont.*

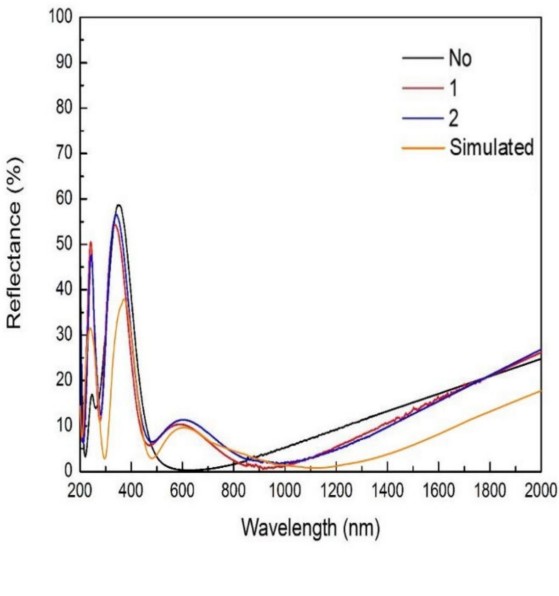

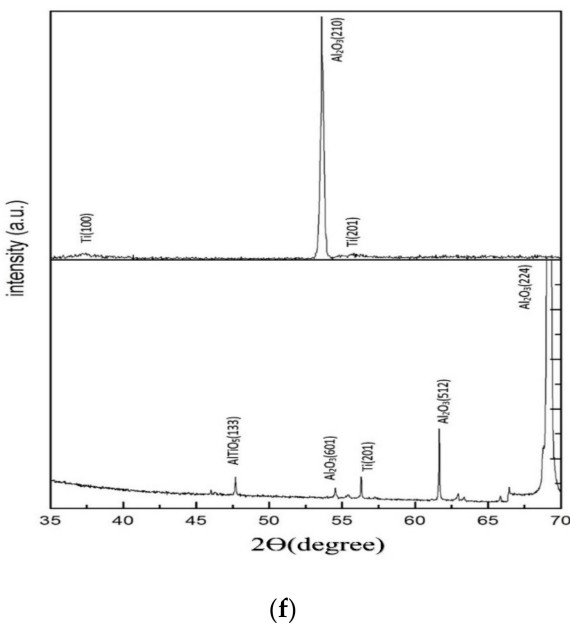

(**e**)                                                                                    (**f**)

**Figure 4.** (**a**) SEM top-view image, (**b**) AFM image and (**c**) SEM cross-sectional image of the $Al_2O_3$/Ti/$Al_2O_3$ CSSA before annealing, and(**d**) SEM top-view image after annealing. (**e**) Optical properties, respectively, before and after annealing 1 and 3 times for 16 h at 550 °C and one simulated optical reflectance. (**f**) X-ray diffraction patterns of the multilayer stack before and after annealing 3 times for 16 h at 550 °C.

## 4. Conclusions

Given that the lifetime of buildings in Eurpoe is more than 200 years, it is important that CSSAs used in building applications must have good color brightness rendering, an excellent photo-thermal conversion efficiency and a long life. However, good color brightness rendering and excellent PTCE seem to be in conflict, so it is necessary to achieve these two advantages through good design. Optical thin film design plays an important role in the coating of the optical thin film. The CUDA C rapid parallel computation technology, optical thin film design software and color distribution analysis programs were used to obtain the optimum parameters for $Al_2O_3$/Ti/$Al_2O_3$ films with both high photo–thermal conversion efficiency, good color brightness rendering and the best color distribution in order to optimize the design for colored solar selective absorbers. It was found that a wide color distribution that included all colors could be obtained for $Al_2O_3$/Ti/$Al_2O_3$ CSSA with a PTCE of more than 90%. With a high PTCE of more than 90% and good color brightness rendering, five colored $Al_2O_3$/Ti/$Al_2O_3$ films for colored solar selective absorbers were designed and fabricated, and their performances were measured and analyzed. The best PTCE of a fabricated CSSA and its thermal emittance efficiency were 93.2% and 7.1%, respectively. There were no changes in the performances observed after annealing three times for 16 h at a temperature of more than 450 °C. The temperature difference between the thermal stability of 450 °C and working temperature of 150 °C was up to 300 °C. The better the temperature difference between thermal stability and working is, the longer the lifetime of the CSSA. Therefore, their lifetimes could be very long. After annealing three times for 16 h at 550 °C, the (210) orientation was transformed to (224) and (512) orientations in the $Al_2O_3$ layer, and the layers were well crystallized with (201) for Ti and (133) for $AlTiO_5$, and no $TiO_2$. Therefore, the Ti layer was not oxidized to $TiO_2$. Ti diffused into the $Al_2O_3$ layer to form a compound $AlTiO_5$ structure. This $AlTiO_5$ layer was very dense with no pinholes. The multilayer structure of the specimen changed from $Al_2O_3$/Ti/$Al_2O_3$ to $Al_2O_3$/$AlTiO_5$/Ti/$Al_2O_3$. The multilayer thickness of the specimen changed from 70/15/87 to 65/14/10.5/87 nm. The PTCE was still more than 91%. Therefore, the $Al_2O_3$/Ti/$Al_2O_3$ films with good performances, including PTCE,

thermal emittance efficiency, structure, adhesion and thermal stability, are good candidates for colored solar selective absorber materials.

**Author Contributions:** Conceptualization, F.-D.L.; Data curation, F.-D.L. and W.-Y.L.; Formal analysis, F.-D.L.; Funding acquisition, F.-D.L.; Investigation, F.-D.L. and W.-Y.L.; Methodology, F.-D.L.; Project administration, F.-D.L.; Software, W.-Y.L.; Supervision, F.-D.L.; Validation, F.-D.L. and W.-Y.L.; Visualization, W.-Y.L.; Writing—original draft, F.-D.L.; Writing—review & editing, F.-D.L. All authors have read and agreed to the published version of the manuscript.

**Funding:** This research is supported by the National Science Council in Taiwan under Contract No. MOST 109-2637-E-992-004.

**Data Availability Statement:** Data sharing is not applicable to this article.

**Acknowledgments:** This work is sustained by the National Science Council in Taiwan under Contract No. MOST 109-2637-E-992-004 -. The authors also want to thank the Taiwan Semiconductor Research Institute for providing the UV–Vis-NIR measurement equipment.

**Conflicts of Interest:** The authors declare no conflict of interest.

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
