# Peer review of "Design, Fabrication and Analysis for Al2O3/Ti/Al2O3 Colored Solar Selective Absorbers for Building Applications"

_coatings, doi:10.3390/coatings12040521_

Round 1
Reviewer 1 Report
The manuscript “Design, Fabrication and Analysis for Al2O3/Ti/Al2O3 Colored Solar 2 Selective Absorbers for Building Applications” is an interesting study. It does fit to the scope of the journal. The following are my comments:
- The use of word ‘clearly’ in line 62 may be avoided.
- Mention the novelty of the study in the last paragraph of the Introduction section.
- The experimental detail section should be in past tense.
- Figure 1 should be improved. There are looking dull.
- Conclusion may be shortened to be more compact.
- Kindly explain in text if it is a hypothetical study or real one.
- The following study may further improve the impact and visibility of the study:
https://doi.org/10.1016/j.nanoen.2021.106834, https://doi.org/10.1016/j.physrep.2021.07.002
- Future scope may added to the conclusion.
- A list of abbreviation should be added.
- A list of symbols should be add
Overall the study is compact and well written. It may be recommended for publication after above mentioned minor revision.
Author Response
The manuscript “Design, Fabrication and Analysis for Al2O3/Ti/Al2O3 Colored Solar 2 Selective Absorbers for Building Applications” is an interesting study. It does fit to the scope of the journal. The following are my comments:
Q1. The use of word ‘clearly’ in line 62 may be avoided.
Ans: “be clearly calculated and analyzed” has been revised to be “be calculated and analyzed in detail”.
Q2. Mention the novelty of the study in the last paragraph of the Introduction section.
Ans:
Yes, “CSSA used in building applications must have good visible brightness.”
“Color rendering“ has been revised to be “Color brightness rendering“.
CSSA used in building applications must have good visible brightness i.e., good color brightness rendering. When the peak value of the optical reflectance in the visible region must be greater than 10%, CSSA will have good color brightness rendering.
Q3. The experimental detail section should be in past tense.
Ans. They have been revised.
Q4. Figure 1 should be improved. There are looking dull.
Ans: Figure 1 has been revised to be concise and clear and easy to be readed.
Q5. Conclusion may be shortened to be more compact.
Ans: Unimportant notes in the conclusion have been removed.
Q6. Kindly explain in text if it is a hypothetical study or real one.
Ans: Figure 3, Figure 4 and Table 1 are the experimental results.
Q7. The following study may further improve the impact and visibility of the study:
https://doi.org/10.1016/j.nanoen.2021.106834, https://doi.org/10.1016/j.physrep.2021.07.002
Ans: They have been added to references 15 and 16.
Q8. Future scope may added to the conclusion.
Ans: Conclusion has been revised.
Q9. A list of abbreviation should be added.
Ans: Abbreviation has been added in page 1.
Q10. A list of symbols should be add.
Ans:
There are no special symbols in this paper.
Putting all symbols below the math makes it easier for the reader to read.
Q11. Overall the study is compact and well written. It may be recommended for publication after above mentioned minor revision.
Ans: Thank you for your helps.
Reviewer 2 Report
This paper is devoted to the development of Al2O3/Ti/Al2O3 thin films as new solar selective absorbers. This material is intended to cover external walls or roof of a building. The coating demonstrates impressive color rendering, an excellent photo-thermal conversion efficiency and a long lifetime.
The work is generally well written. The comments mainly refer to drawings that are not sharp 1 (d-g), 2, 3, 4 (e-f) and therefore are poorly “readable”. In addition, the authors should take care that all the figures would clearly indicate the values that are plotted along the axes and the units of measurement (definite or arbitrary units) are given.
In addition, it would be important for practical application of the described coating to compare it with existing analogues in terms of cost. I would like to see such information in this manuscript.
Author Response
This paper is devoted to the development of Al2O3/Ti/Al2O3 thin films as new solar selective absorbers. This material is intended to cover external walls or roof of a building. The coating demonstrates impressive color rendering, an excellent photo-thermal conversion efficiency and a long lifetime.
Q1. The work is generally well written. The comments mainly refer to drawings that are not sharp 1 (d-g), 2, 3, 4 (e-f) and therefore are poorly “readable”.
Ans: Fig.1-4 all have been revised to be easily readable.
Q2. In addition, the authors should take care that all the figures would clearly indicate the values that are plotted along the axes and the units of measurement (definite or arbitrary units) are given.
Ans: All Figures have been revised.
Q3. In addition, it would be important for practical application of the described coating to compare it with existing analogues in terms of cost. I would like to see such information in this manuscript.
Ans:
Since there are only four layers in the film and they are very thin, the process is fast and then the cost is cheap. Therefore, in terms of cost, it could be cheaper than other CSSAs.
Reviewer 3 Report
In this research work, authors have claimed to have a good colored solar collective absorber, and they have claimed for obtaining 99% photo-thermal conversion efficiency (PTCE) of the fabricated absorber. After critically reviewing the manuscript, authors are encouraged to answer the following
comments/questions:
1. It is hard to find enough relevant research work related to the main material of the proposed absorber (Al2O3/Ti/Al2O3); therefore, it is highly recommended to revise the Introduction of manuscript as per the following aspect:
Literature should be comprised of the properties of the above-mentioned material and authors are recommended to relate those properties with the solar collective absorber.
2. A lot of research has been also been observed primarily in the utilization of different materials for color rendering as well as enhancement in the PTCE of the solar collective absorber. Therefore, how can we defend that the fabricated absorber (based on Al2O3/Ti/Al2O3) is best fitted for building applications?
3. Authors are encouraged to enlist the possible reasons for obtaining the highest PTCE for sample e (Primarily in context with sample d)?
4. Why authors have only considered sample ‘e’ for the comparison between measured and simulation values?
5. It is recommended to paraphrase section 3.2.1 in the context of the fabrication of each sample rather than explaining it generally as it would be easy for the readers to distinguish between each prepared sample.
6. Figure 4 should be revised completely as it is quite difficult to analyze the X-ray diffraction patterns.
7. In the highlighted sentence, the authors have indirectly claimed that if peak values are greater than ~10 % then the appearance of the prepared samples shall be clear. “The peak values of the measured optical reflectance in the visible region for five fabricated CSSA are all greater than 10%, therefore, the appearance of these colors in the sunlight is very clear and bright”
Can the authors justify this claim? On the contrary, it is recommended to cite the reference article in the statement.
8. Many typos are in the manuscript. Authors are requested to carefully proofread the manuscript.

Author Response
In this research work, authors have claimed to have a good colored solar collective absorber, and they have claimed for obtaining 99% photo-thermal conversion efficiency (PTCE) of the fabricated absorber. After critically reviewing the manuscript, authors are encouraged to answer the following
comments/questions:
Q1. It is hard to find enough relevant research work related to the main material of the proposed absorber (Al2O3/Ti/Al2O3); therefore, it is highly recommended to revise the Introduction of manuscript as per the following aspect:
Literature should be comprised of the properties of the above-mentioned material and authors are recommended to relate those properties with the solar collective absorber.
Ans:
It is revised to be “The simple experimental schematic for the absorber is shown in Figure 1(c). Al2O3 layer is a transparent layer without any absorption function. The middle Ti layer is an absorption layer. The Ti substrate has two functions: reflection and absorption. At the interfaces of 1 to 4 in Figure 1(c), an incident light will be partially refracted and partially reflected. Therefore, the absorber is the Al2O3/Ti/Al2O3/Ti film.”
Q2. A lot of research has been also been observed primarily in the utilization of different materials for color rendering as well as enhancement in the PTCE of the solar collective absorber. Therefore, how can we defend that the fabricated absorber (based on Al2O3/Ti/Al2O3) is best fitted for building applications?
Ans:
A good solar selective absorbers (SSA) used in building applications must have three good performances: a good color brightness rendering, an excellent photo-thermal conversion efficiency (PTCE) and a long life. Our fabricated absorbers (based on Al2O3/Ti/Al2O3) have the three good performances listed above, therefore, it is fitted for building applications.
Q3. Authors are encouraged to enlist the possible reasons for obtaining the highest PTCE for sample e (Primarily in context with sample d)?
Ans:
Sample e is just the best piece of PTCE among the samples produced in this laboratory.
Q4. Why authors have only considered sample ‘e’ for the comparison between measured and simulation values?
Ans:
Sample e is the easiest to be made in five color samples. And in 3.2.2 Performance measurements and analysis of fabricated Al2O3/Ti/Al2O3 solar selective absorbers, samples similar to Sample e were used for measurement and analysis. The purpose of sample e with the simulated curve is to demonstrate that the optical performance results of the designed film are very close to the measured results of the coated film. In order to avoid complexity in Figure 3, only sample e has simulated curve.
Q5. It is recommended to paraphrase section 3.2.1 in the context of the fabrication of each sample rather than explaining it generally as it would be easy for the readers to distinguish between each prepared sample.
Ans:
Layer thickness of Al2O3/Ti/Al2O3 and chromaticity coordinate have been added to Table 1. And Table 1 has been revised to be as follows:
Table 1. Layer thickness of Al2O3/Ti/Al2O3, chromaticity coordinate, PTCEs and εs of the fabricated samples (a)-(e) in Fig. 3.
Sample |
Layer Thickness of Al2O3/Ti/Al2O3 (nm) |
Chromaticity Coordinate |
PTCE (%) |
ε (%) |
a |
45/15/50 |
(0.410, 0.227) |
92.3 |
7.6 |
b |
65/15/70 |
(0.375, 0.467) |
90.5 |
9.1 |
c |
60/15/69 |
(0.2861, 0.469) |
91.1 |
8.5 |
d |
55/15/69 |
(0.184, 0.348) |
93.0 |
7.3 |
e |
50/19/58 |
(0.168, 0.064) |
93.2 |
7.1 |
Q6. Figure 4 should be revised completely as it is quite difficult to analyze the X-ray diffraction patterns.
Ans: It has been revised.
Q7. In the highlighted sentence, the authors have indirectly claimed that if peak values are greater than ~10 % then the appearance of the prepared samples shall be clear. “The peak values of the measured optical reflectance in the visible region for five fabricated CSSA are all greater than 10%, therefore, the appearance of these colors in the sunlight is very clear and bright” Can the authors justify this claim? On the contrary, it is recommended to cite the reference article in the statement.
Ans:
The peak values of optical reflectances less than 10% in the visible region in CSSAs of references [21-26] still have good color brightness rendering. Therefore, our five CSSAs have good color brightness rendering.
Q8. Many typos are in the manuscript. Authors are requested to carefully proofread the manuscript.
Ans: They have been revised.
Round 2
Reviewer 3 Report
The authors have modified the manuscript significantly and answered all the questions of reviewers. The manuscript is in good form and can be publish in current form.